# The High Content of *Ent-*11α-hydroxy-15-oxo-kaur- 16-en-19-oic Acid in *Adenostemma lavenia* (L.) O. Kuntze Leaf Extract: With Preliminary in Vivo Assays

**DOI:** 10.3390/foods9010073

**Published:** 2020-01-09

**Authors:** Akie Hamamoto, Ryosuke Isogai, Miwa Maeda, Masumi Hayazaki, Eito Horiyama, Shigeo Takashima, Mamoru Koketsu, Hiroshi Takemori

**Affiliations:** Department of Chemistry and Biomolecular Science, Faculty of Engineering, Gifu University, 1-1 Yanagido, Gifu 501-1193, Japan; ahama@gifu-u.ac.jp (A.H.); y4521005@edu.gifu-u.ac.jp (R.I.); v3032132@edu.gifu-u.ac.jp (M.M.); v3032126@edu.gifu-u.ac.jp (M.H.); y4524067@edu.gifu-u.ac.jp (E.H.); staka@gifu-u.ac.jp (S.T.); koketsu@gifu-u.ac.jp (M.K.)

**Keywords:** 11α-hydroxy-15-oxo-kaur-16-en-19-oic acid, anti-melanogenesis, skin whitening, functional food, medical plant, *Adenostemma lavenia*, *Pteris dispar* Kunze, tyrosinase, B16F10 melanoma

## Abstract

Background: *Ent-*11α-hydroxy-15-oxo-kaur-16-en-19-oic acid (11αOH-KA) is a multifunctional biochemical found in some ferns, *Pteris semipinnata*, and its congeneric species. Although a number of therapeutic applications of 11αOH-KA have been proposed (e.g., anti-cancer, anti-inflammation, and skin whitening), the content of 11αOH-KA in these ferns is not high. *Adenostemma lavenia* (L.) O. Kuntze, an Asteraceae, has also been reported to contain 11αOH-KA. The decoction (hot water extract) of whole plants of *A. lavenia* is used as a folk remedy for inflammatory disorders, such as hepatitis and pneumonia, suggesting that 11αOH-KA may be the ingredient responsible for the medicinal properties of this plant. Methods: The anti-melanogenic activities of the water extracts of *A. lavenia* leaves and *Pteris dispar* Kunze (a cognate of *P. semipinnata*) leaves were compared in mouse B16F10 melanoma cells. The amount of 11αOH-KA was measured by using liquid chromatography spectrometry. C57BL/6J mice were treated with the water extract of *A. lavenia* leaf, and the blood concentration of 11αOH-KA was measured. The in vivo efficacy of the water extract of *A. lavenia* leaf was evaluated according to tis anti-melanogenic activity by monitoring hair color. Results: Although both the extracts (*A. lavenia* and *P. dispar* Kunze) showed high anti-melanogenic activities, only *A. lavenia* contained a high amount of 11αOH-KA, approximately 2.5% of the dry leaf weight. 11αOH-KA can be purified from *A. lavenia* leaves in two steps: water extraction followed by chloroform distribution. The treatment of mice with the water extract of *A. lavenia* leaf suppresses pigmentation in their hairs. Conclusions: Despite the small number of mice examined, the present preliminary result of the suppressed hair pigmentation suggests that the water extract of *A. lavenia* leaf and the ingredient that is possibly responsible for this—11αOH-KA—are new materials for oral cosmetics. The results may also be helpful in the future development of functional foods and methods to treat patients suffering from hyperpigmentation disorders, such as melasma.

## 1. Introduction

Studies on anti-melanogenic reagents for skin whitening and anti-aging have become more frequent, especially for ingredients in foods (fruits, vegetables, and medical herbs) [1]. A number of polyphenols and terpenes and their sources have been developed and commercialized as functional foods and supplements.

The diterpene/kaurene *ent-*11α-hydroxy-15-oxo-kaur-16-en-19-oic acid (11αOH-KA), found in the leaves of ferns (*Pteris semipinnata* [2,3,4,5], *Pteris livida* [6]) and Compositae (*Gochnatia decora* [7]), is also an attractive candidate for anti-melanogenic reagents [5]. Despite its broad physiological activities (e.g., anti-cancer and anti-inflammation) [8,9,10,11], there are only a few records on the in vivo efficacy of 11αOH-KA, including its anti-melanogenic potential, likely due to its rarity and restricted number of sources.

Structure–activity relationship studies, both in vitro and in cultured cells, have identified the moieties in 11αOH-KA important for exerting its functional activities (Figure 1). Replacement of the carboxy group at the 19th position in 11αOH-KA by methyl or hydroxy groups converts this compound to a compound with high anti-cancer activities, which indicates that the 19-carboxyl group weakens the cellular toxicity of this compound [5,10,12]. In contrast, the alkene group at the 16th position, the ketone group at the 15th position, and the hydroxy group at the 11α position enhance the apoptotic activities by activating the Caspase signaling pathway [10,12]. However, these moieties are not indispensable for the cell death signals of kaurenes. The structural requirements for anti-inflammatory activity are similar to those for anti-cancer activity, suggesting crosstalk via common cellular targets, such as nuclear factor kappa B (NF-κB) [12].

Tyrosinase is a rate-limiting enzyme in melanin synthesis [13,14]. The enzyme catalyzes sequential reactions: the conversion of tyrosine to *l*-3,4-dihydroxyphenylalanine, *l*-Dopa, and then to dopaquinone. We have reported that 11αOH-KA inhibits the expression of the *tyrosinase* gene, which results in a suppression of melanogenesis in B16F10 mouse melanoma cells [5]. The moieties of 11α-hydroxy, 15-oxo, and 16-en in 11αOH-KA are indispensable for its anti-melanogenesis activity, which is a different characteristic of the structural requirements of 11αOH-KA compared to other activities, such as anti-cancer or anti-inflammatory activities.

Thus, in terms of its anti-melanogenic activity, 11αOH-KA is a unique compound among kaurenes. Unfortunately, *Pteris dispar* Kunze, our previous source of 11αOH-KA, contained only a small amount of this compound [5], and this fern, as well as *P. semipinnata* (a cognate) [4,15], has no record of usage in human subjects. Moreover, in all past experiments, organic solvents (methyl or ethyl alcohol, chloroform, or hexane) were used to extract 11αOH-KA, without any records of the usage of aqueous extraction.

Almost 40 years ago, an American group discovered 11αOH-KA and its derivatives in *Adenostemma lavenia* (L.) O. Kuntze (an Asteraceae) [16]. Although *A. lavenia* has not been studied sufficiently regarding the risk of possible contaminants (e.g., inorganic pollutants [17] and secondary metabolites produced by fungi [18,19]) in leaves and other parts, whole plant powder (named Ma Zhi Hu) is sold in the United States under the permission of medical doctors (the products can be obtained from QualiHerb, Cerritos, CA, USA: Appendix A). Therefore, in this study, by measuring the 11αOH-KA content and evaluating the efficacy of its anti-melanogenic activities, we examined the potential of a water extract of *A. lavenia* leaf for use in internal cosmetics, as well as a new category of functional foods [20,21,22] that may be used to treat patients suffering from hyperpigmentation disorders such as melasma. To date, only tranexamic acid (a hemostatic drug) has been available as an internal drug for hyperpigmentation disorders [23,24].

We report here that *A. lavenia* leaves contain 11αOH-KA at high levels (approximately 2.5% of the dry weight of leaves). The water extract of the *A. lavenia* leaf efficiently suppresses melanogenesis in B16F10 cells, and 11αOH-KA may be responsible for 50% of the anti-melanogenic activities of the water extract. The oral treatment of mice with the water extract of *A. lavenia* leaf results in the suppression of their hair pigmentation.

## 2. Materials and Methods

### 2.1. Propagation of A. lavenia

Since *A. lavenia* is listed as an endangered species in most areas of Japan, we cultured this plant in a field near our laboratory. *A. lavenia* was propagated by division from 10 seedlings, and almost 100 g of dried leaves harvested from June to July 2019 was used in this study.

### 2.2. Purification of 11αOH-KA from Water Extracts and Its Analysis

Pure 11α-OH KA (authentic) was purchased from BioBioPha (Shanghai, China). Dried leaf powder (100 mg) obtained from the leaves of *A. lavenia* or *P. dispar* Kunze was soaked in 1 mL of distilled water (for a 10-volume extraction) for 12 h, and the residuals were removed by centrifugation at 12,000 rpm for 10 min. The resulting extracts were analyzed by high-performance (HP) liquid chromatography (LC) using a C_18_ ODS column (4.6 mm × 50 mm: RP-18 GP, Kanto Kagaku, Tokyo, Japan) with a gradient ranging from 40% methanol/water to 100% methanol over 10 min. The 11α-OH KA detection was performed by determining the ultraviolet (UV) absorption at 245 nm (A_245_). For 20-, 30-, and 50-volume extractions, 50 mg, 33 mg, and 20 mg of dried leaf powder were extracted by 1 mL of distilled water.

To confirm the structure of 11α-OH KA in *A. lavenia*, kaurenoic acid was purified at a semi-large scale. A total of 100 g of dried *A. lavenia* leaves was extracted twice from 20 volumes of water, an equi-volume of CHCl_3_ was added to the water extract of the leaves, and 11α-OH KA was recovered from the CHCl_3_ phase. During the evaporation of the recovered CHCl_3_ phase, 11α-OH KA was precipitated as white crystals. These crystals were recovered by a paper filter. The recovered crystals were analyzed by nuclear magnetic resonance (NMR) (^1^H-NMR (CDCl_3_) δ 0.95 (3H, s), 1.28 (3H, s), 1.05–2.22 (15H, m), 2.37 (1H, d, J = 11.9 Hz), 3.06 (1H, broad s), 4.06 (1H, d, J = 4.6 Hz), 5.27 (1H, s), and 5.87 (1H, s)). The ^1^H-NMR spectrum of the crystal was coincident with the authentic 11α-OH KA spectrum (see Appendix A).

### 2.3. Analysis of 11αOH-KA in Mouse Serum

Methods for the ultra-performance LC (UPLC)-mass spectrometry (MS) analyses were drawn from several methods used in [25,26]. Briefly, to treat mice (C57BL/6J), *A. lavenia* leaves were extracted with 30 volumes of water. After sterilization at 120 °C for 20 min, the extract was further diluted with 2 volumes of water and served as drinking water. To examine the pharmacokinetics of 11α-OH KA, 100 mg/kg of the compound suspended in 1% Tween 80 (Kanto Kagaku, Tokyo, Japan) was administered orally. Blood was recovered via the tail vein 15–240 min after administration.

To measure the 11α-OH KA in mouse blood, the serum (20 µL) was extracted with 200 µL of acetonitrile. To normalize the extraction efficiency, 10 nmols of 18β-glycyrrhetinic acid (Kanto Kagaku) was added to the serum prior to the extraction. After the removal of acetonitrile by evaporation, the compounds were dissolved in 5 µL of methanol followed by the addition of 5 µL of H_2_O. Then, 1 µL of solution was used for LC-MS spectrometry (MS: Waters Xevo G2-XS QTof, Waters, Milford, MA). To separate 11α-OH KA, a C_18_ ODS column (3.0 mm × 150 mm (5 µm): RP-18 GP, Kanto Kagaku) and a gradient ranging from 35% acetonitrile/water to 100% acetonitrile over 10 min were applied. The MS conditions were as follows: capillary (kV), 2.20; sampling core, 45; extraction cone, 4.0. Because we failed to detect fragments of 11α-OH KA via MS/MS analyses (Appendix A), 11α-OH KA was quantified with only MS signals by monitoring the mass-to-charge ratio (*m/z*) of 331.1879 with the 0.01 Da path (Appendix A). A standard curve was prepared by the addition of 11α-OH KA (at a final concentration of 0.1–3 µM) to 50% methanol/water that had been prepared from normal mouse serum using the same procedure for 11α-OH KA extraction. The 18β-glycyrrhetinic acid was measured using LC-UV under the same conditions used for the 11α-OH KA measurement. These animal experiments were approved by the Animal Experiment Committee of Gifu University (H30-041). We used a minimum number of mice to comply with animal welfare requirements.

### 2.4. Melanogenesis Assay

Mouse B16F10 melanoma cells from the American Type Culture Collection (RIKEN, Tsukuba, Japan) were cultured in Dulbecco’s modified Eagle’s medium (DMEM-High glucose (4.5 g/L d*-*glucose), WAKO, Kyoto, Japan) and supplemented with 10% fetal bovine serum and antibiotics. The cells were incubated in a humidified atmosphere with 5% CO_2_ at 37 °C and were transferred every two days via trypsinization to new culture plates.

The cells (2 × 10^5^) were plated onto a 6-well plate and incubated for 12–16 h prior to the induction of melanogenesis. Melanogenesis was induced by 20 µM forskolin (Fsk) together with *A. lavenia* water extract or 11α-OH KA for 48 h. Melanin measurements were performed in accordance with the literature [5]. Briefly, the cells were washed twice with phosphate-buffered saline and recovered in 2 mL collection tubes, followed by centrifugation at 8000 rpm for 1.5 min. Each cell pellet was suspended in 200 μL of distilled water, and 50 μL was used for the measurement of protein content. The residual 150 μL was mixed with 150 μL of 2 M NaOH and then lysed by incubation at 45 °C for 2 h. The melanin was extracted with a 2:1 chloroform–methanol mixture and measured with a spectrophotometer (Glomax Multi, Promega, Madison, WI, USA) at 405 nm. The protein concentration of the cell pellets was determined using a Bradford reagent (Bio-Rad, Hercules, CA, USA) and used for normalization of the melanin content.

### 2.5. Statistical Analysis

The statistical analysis was performed using Student’s *t* test when the data were obtained during the same experiment. We did not use statistical analyses if the experiments were performed on different days or if the sample number was only two.

## 3. Results

### 3.1. A. lavenia Contains a High Amount of 11αOH-KA

We previously reported that the ethanol extract of *P. disper* Kunze leaves, prepared with 10 volumes of dry weight (*v*/*w*) ethanol, efficiently inhibited melanogenesis in B16F10 cells, even after a 1/1000 dilution [5]. To compare the efficacy of the water extracts of *A. lavenia* dried leaves and that of the water extracts of *P. disper* Kunze leaves, we prepared the extracts (20 volumes of water (*v*/*w* dried leaves)) and added them into the cultured medium together with forskolin (melanogenesis inducer—a cAMP agonist). Both extracts efficiently inhibited melanin synthesis even at 1/1000 dilution, when the melanin content was visually evaluated (Figure 2A). The melanin content was measured after NaOH extraction (Figure 2B). At 1/3000 dilution, the *A. lavenia* extract decreased the melanin content by 69% compared to the control (Fsk only), and the *P. disper* Kunze extract decreased the melanin content by 41%. At 1/1000 dilution, *A. lavenia* extract decreased the melanin content completely, while the *P. disper* Kunze extract decreased the melanin content by 60%. This difference was not observed when the extracts were diluted at 1/300.

The LC-UV analyses suggested that 11αOH-KA was the major component in the *A. lavenia* water extract (Figure 2C, left). In contrast, no clear peak for 11αOH-KA was detected when the *P. disper* Kunze extract was analyzed (Figure 2C, right). The LC-MS analyses suggested that the peak indicated by an arrow corresponded to 11αOH-KA (data not shown). These results indicate that 11αOH-KA might be the major component responsible for melanogenesis inhibition in *A. lavenia* extract, while derivatives like glycosylated forms [5] might preferentially contribute to melanin inhibition in *P. disper* Kunze extract.

Next, we tried to purify 11αOH-KA by distribution. CHCl_3_ was able to extract 11αOH-KA from the above *A. lavenia* water extract with a high purity (more than 87%) (Figure 2D, left). The spectra of the NMR analyses of the crystallized substance in CHCl_3_ were identical to those of the authentic 11αOH-KA (see Materials and Methods). In contrast, a large number of unknown compounds were extracted from the *P. disper* Kunze water extract (Figure 2D, right). A summary of the LC analyses is indicated in Figure 2E, suggesting that *A. lavenia* is an attractive source of 11αOH-KA.

### 3.2. Suitable Extraction of 11αOH-KA from A. lavenia

According to information found on the Internet, all portions (leaf, stem, and root) of *A. lavenia* are used as herbal medicines in China and Taiwan. To determine the best conditions for 11αOH-KA extraction from *A. lavenia,* we first examined which portions of the plant are rich in this compound. Water extracts were prepared from leaves alone or from aerial parts. The melanogenesis assay on the B16F10 cells showed that the extract prepared from the aerial parts (stems and leaves) had less anti-melanogenic activity than that prepared from the leaves alone (Figure 3A, left). This suggests that stems (which account for more than 90% of the dry weight) contain less 11αOH-KA, which was confirmed by the LC-UV analyses (Figure 3A, right).

Next, we examined how many volumes of water should be used for the extraction. Apparently, a small volume of water (10-fold) could enrich 11αOH-KA (Figure 3B). We speculated that the low solubility of 11αOH-KA in water might help retain this compound in residues. A maximum recovery of 11αOH-KA in water was observed when more than 30 volumes of water were used (Figure 3C). Therefore, considering the maximum recovery and enrichment of 11αOH-KA in water, we used 30 volumes of water in later experiments.

In China and Taiwan, *A. lavenia* is extracted in hot water, through which the thermal stability of 11αOH-KA was examined. The water extracts were incubated at 90 °C or 120 °C for 10 or 20 min, and the 11αOH-KA content was analyzed by LC-UV. These thermal treatments produced a large amount of sedimentation, likely consisting of denatured protein. The 11αOH-KA contents in the soluble fraction were apparently increased (Figure 3D), which might be a result of the removal of the sedimented materials.

#### 3.2.1. αOH-KA Is the Compound in *A. lavenia* Water Extract Responsible for Anti-Melanogenic Activity in B16F10 Cells

When 1 g of leaf was extracted in 30 volumes of water, the content of 11αOH-KA was approximately 0.6 mg/mL (1.8 mM). To determine the degree of the contribution of 11αOH-KA in the *A. lavenia* water extract to the anti-melanogenic activity, different concentrations of authentic 11αOH-KA were tested in B16F10 cells. Visual examination (Figure 4A) and the measurement of melanin content (Figure 4B) revealed that the water extracts (Figure 4A, upper set) that showed the same efficacy in terms of anti-melanogenic activity as authentic 11αOH-KA alone (Figure 4A, lower set) contained only a half amount of 11αOH-KA. These results suggest that the 11αOH-KA in the extract contributed to almost 50% of the extract’s anti-melanogenic activity. The other 50% might be derived from 11αOH-KA derivatives, such as glycoside forms, which might also explain the low content of 11αOH-KA in the *P. disper* Kunze extract despite its high efficacy in terms of anti-melanogenic activity (Figure 2).

#### 3.2.2. αOH-KA Is the Compound in *A. lavenia* Extract Responsible for Anti-Melanogenic Activity in Mice

To obtain preliminary results for the future development of *A. lavenia* as a material for cosmetics, we performed in vivo analyses using a small number of mice. We first examined the pharmacokinetics of 11αOH-KA in mice. The water intake was approximately 4 mL daily in mice with approximately 25 g bodyweight. The maximum solubility of the 11αOH-KA in the water extract of *A. lavenia* (see Figure 3C, 20–30 volumes water: 0.6 mg/mL) could theoretically provide a 100 mg/kg daily dose of this kaurenoic acid (4 mL of 0.6 mg/mL 11αOH-KA made 2.4 mg/day; normalization of 2.4 mg by mouse body weight (25 g) resulted in 96 mg/kg). Therefore, we decided to treat the mice with 100 mg/kg 11αOH-KA (suspended in 200 µL of 1% Tween 80).

Oral treatment with 11αOH-KA resulted in its rapid appearance (within 15 min) in the blood (Figure 5A). The concentration reached and retained an effective dose in the B16F10 cells. This result indicates that 11αOH-KA could be absorbed through the digestive tract. However, the fact that the concentration decreased within 60 min suggests rapid excretion of this compound.

Next, we examined the chronic effects of 11αOH-KA in mice. Since even 0.6 mg/mL of 11αOH-KA rapidly precipitated in drinking water if 11αOH-KA was pure, we decided to test the *A. lavenia* water extract. However, the *A. lavenia* water extract might contain unknown ingredients with a bitter taste. Therefore, the extract had to be further diluted with two volumes of water (with a final value of 0.3 mg/mL for 11αOH-KA, approximately 50 mg/kg/day). The mice were treated with the water extracts for 1 week, and their blood samples were then collected (Figure 5B). LC-MS analyses detected 11αOH-KA in the blood at a semi-effective concentration. The *P. dispar* Kunze water extract that had been prepared by the same procedure as the *A. lavenia* water extract was also examined. The content of 11αOH-KA in the blood of mice treated with the *P. dispar* Kunze extract was 10 times lower than that of the mice treated with the *A. lavenia* extract.

In adult mice, hair growth pauses during the telogen stage for several months, and new growth (anagen) randomly occurs during only a few days in different areas [27]. Therefore, to examine the effect of melanogenesis inhibitors, the duration of the treatment with compounds (inhibitors) should be long (e.g., several months). In contrast, downy hairs are synchronously replaced with adult hairs during the 3–4-week-old (W) phase. Therefore, when 3W mice are treated with inhibitors for 2 weeks, the effects on melanogenesis can be evaluated from the pigmentation (color) of the newly grown hair (i.e., the first adult hair) at 5W–6W [28].

When mice were treated with *A. lavenia* extract via drinking water for two weeks (3W–4W), their pigmentation was suppressed in the newly grown hairs (Figure 5C). In contrast, their eye pigmentation was not affected. The suppression of hair pigmentation was not observed in the mice treated with the *P. dispar* Kunze extract. These results suggest that the *A. lavenia* leaf extract might contain anti-melanogenic substances, possibly 11αOH-KA, that are effective in vivo, whereas the *P. dispar* Kunze leaf extract, likely rich in 11αOH-KA derivatives, might not be effective in vivo.

## 4. Discussion

We found that *A. lavenia* leaves contained a high amount of 11αOH-KA (approximately 2.5% of dry leaf weight). Water efficiently extracted 11αOH-KA, despite the hydrophobic nature of this compound. As 11αOH-KA in water was stable at 120 °C for 20 min, we sterilized the water extract of *A. lavenia* leaf and administered it to mice. The results confirmed the presence of 11αOH-KA in the mice’s blood and the potency of the water extract as an anti-melanogenic reagent in vivo.

The presence of 11αOH-KA and its derivatives in *A. lavenia* was reported in 1979 [16] and 1990 [29]. Since then, only one study from Indonesia has reported a comprehensive analysis of the ingredients in *A. lavenia* [30]. However, no 11αOH-KA was identified in those materials. Moreover, no studies on the physical properties of 11αOH-KA, which can be extracted by pure water from *A. lavenia*, have been found.

A review article about folk remedies in India refers to *A. lavenia* water extract as a treatment for digestive system disorders [31]. An Internet search for “*A. lavenia*” or its Chinese name suggested some beneficial uses of this plant as an anti-inflammatory herbal medicine via a hot water extract (i.e., as a decoction) (e.g., for hepatitis, pulmonary inflammation (allergy and chronic obstructive pulmonary disease), and skin inflammation). However, no formal reports on the biological activities of *A. lavenia* extracts or the ecological status of this plant were found in PubMed. In our city (Gifu, Japan), *A. lavenia* is listed as an endangered species. The rarity of *A. lavenia* may make it difficult to analyze and develop its potential as an herbal medicine and a functional food.

On the other hand, the beneficial potential of 11αOH-KA, purified from *P. semipinnata*, has been determined in vitro and in vivo [8,9,10,11]. Although we did not examine the content of 11αOH-KA in *P. semipinnata*, we guess that this content may not be very high compared with that in *A. lavenia* because *P. semipinnata* is closely related to *P. disper* Kunze. In addition, there are no records on the use of these ferns in human subjects, even on the Internet. This background suggests that *A. lavenia* may have greater potential as a herbal medicine than these ferns if 11αOH-KA is shown to have clinical potential.

Studies on the potential of 11αOH-KA have been focused on its activities as an anti-cancer [2,10,11,32] and anti-inflammatory [11,33] agent. However, it is also true that the doses of 11αOH-KA required for these activities are more than 20 µM, which is relatively high. In contrast, at a concentration of less than 1 µM, the kaurene *ent-*11α-hydroxy-16-kauren-15-one, which has a methyl group in the 19-position instead of the carboxyl group in 11αOH-KA, shows high apoptotic activity against human HL-60 monocytes [12]. These differences suggest that the 19-position may be a determinant of cellular toxicity [5].

Although anti-tumor and anti-inflammatory activities are found in a variety of kaurens [7,34,35,36], the specific feature of 11αOH-KA is a type of anti-melanogenic activity [5]. A number of natural compounds, such as flavonoids and terpenoids, have been found to modulate melanogenesis by altering the expression of microphthalmia transcription factor (MITF), which is essential for melanogenic programs [13,14]. In contrast to these compounds, no MITF modulators have been found in kaurens or kaurenoic acids.

11αOH-KA suppresses the expression of the *tyrosinase* gene by downregulating its promoter activity. Although 11αOH-KA does not alter the expression or activity levels of MITF, 11αOH-KA suppresses *tyrosinase* gene promoter activity in an MITF-binding-element-dependent manner [5]. Interestingly, the overexpression of MITF in B16F10 cells results in a cancellation of the 11αOH-KA-mediated suppression of *tyrosinase* promoter activity, suggesting that 11αOH-KA may recruit an unknown repressor to the MITF-binding element. Moreover, the mechanism by which 11αOH-KA suppresses *tyrosinase* promoter activity may be different from that which leads to cytotoxicity because anti-melanogenic activity is observed even when B16F10 cells are treated with 1.2 µM of 11αOH-KA (Figure 4).

Research involving a human three-dimensional skin model composed of primary keratinocytes and melanocytes suggests that the topical use of 1 mM of 11αOH-KA completely suppresses skin pigmentation [5]. In terms of anti-melanogenic activity, 1 mM of 11αOH-KA is equivalent to approximately 6.6 mg/mL (0.66%) of dried *A. lavenia* leaf (Figure 3). In this context, *A. lavenia* is an attractive source of cosmetic items for skin whitening. In addition, *A. lavenia* is used as a folk medicine, and 11αOH-KA has a wide margin between its effective doses (concentrations) for anti-melanogenic activity and cytotoxicity.

Importantly, tranexamic acid (a hemostatic drug) has been found to suppress melanin deposition in human skin and is used as an oral medicine to treat melasma [23,24]. Interestingly, the fur of mice treated with tranexamic acid changed color from black to brown [37]. In the present study, we have also shown that a water extract of *A. lavenia* leaf can efficiently suppress hair pigmentation (black to brown) in mice when given orally, suggesting that *A. lavenia* leaf could be developed into both internal cosmetics and functional foods [20,21,22] for patients suffering from melasma. Further studies are needed regarding the cellular targets of *A. lavenia* extracts and 11αOH-KA, as well as their safety.

## Figures and Tables

**Figure 1 foods-09-00073-f001:**
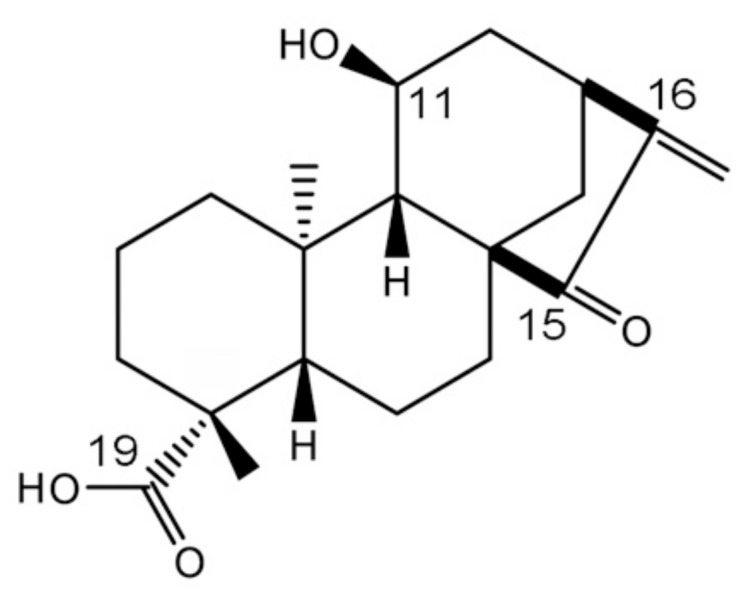
The structure of *ent-*11α-hydroxy-15-oxo-kaur-16-en-19-oic acid (11αOH-KA).

**Figure 2 foods-09-00073-f002:**
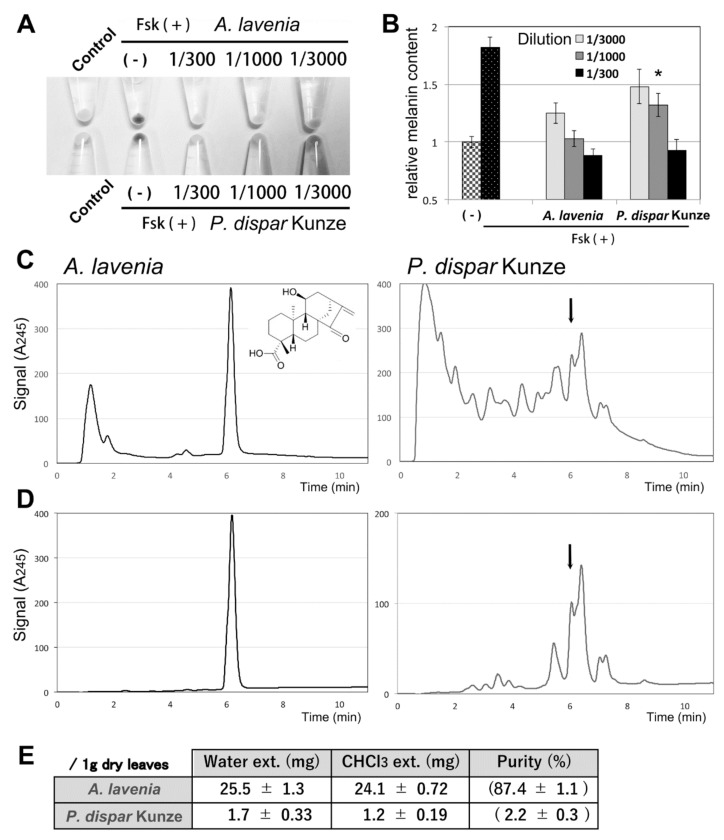
(**A**) B16F10 cells were treated with water extracts of *A. lavenia* and *P. dispar* Kunze (prepared with 20 volumes of water) at the indicated fold dilution (1/300, 1/1000, 1/3000) in the medium. When the water extracts were dried, 1 mL extracts from *A. lavenia* and *P. dispar* Kunze dried leaves (50 mg each) produced approximately 9 mg and 12 mg of precipitants, respectively. Thus, the 1/1000 dilution produced 9 µg/mL and 12 µg/mL, respectively. Melanogenesis was induced by 20 µM of forskolin (Fsk) for 48 h. (**B**) Melanin content was measured at OD_405_ after NaOH extraction. The values were first normalized by the total protein, and the relative content in the control (without forskolin) was set to 1. The means and SD are indicated, *n* = 3. * indicates *p < 0.05* between *A. lavenia* and *P. disper* Kunze. (**C**) LC-UV analyses of the water extracts. An arrow indicates a peak for the candidate of 11αOH-KA that was detected by LC-MS (see Materials and Methods). (**D**) LC-UV analyses of water extracts after purification with 1 volume of CHCl_3_. (**E**) A summary of 11αOH-KA content in the fractions of water extracts (prepared from 1 g leaves with 2 times 20 volumes of water) and those followed by CHCl_3_ extraction. The purity of 11αOH-KA in the final fractions (water followed by CHCl_3_) was calculated from the total dry weight of the final fractions (27.5 mg and 54.5 mg, respectively).

**Figure 3 foods-09-00073-f003:**
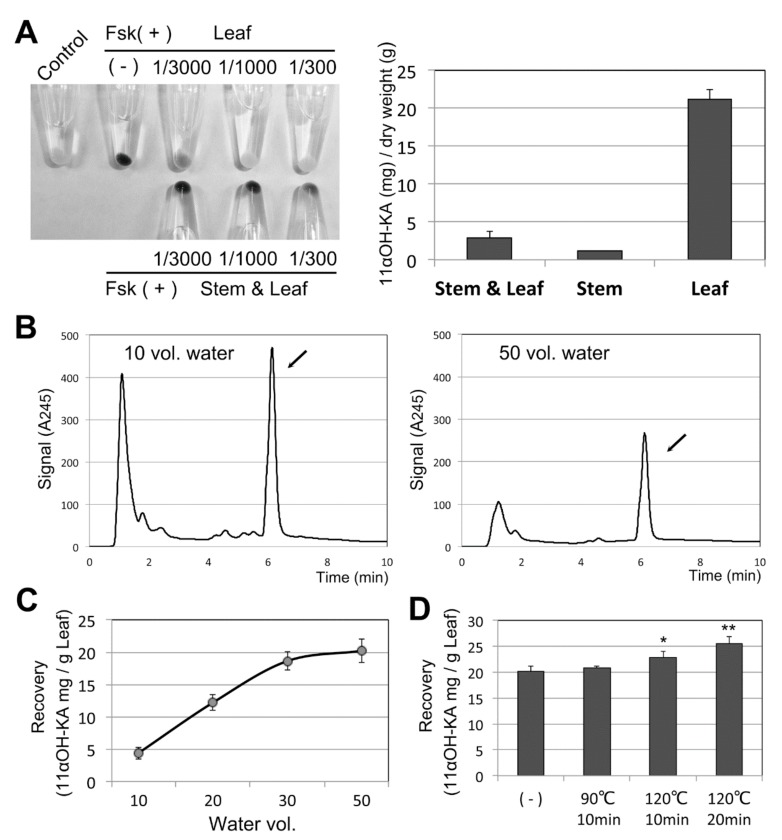
(**A**) Leaves alone and with stems were dried and extracted with 20 volumes of water. B16F10 cells were treated with these extracts at the indicated dilution (left). One gram of dried *A. lavenia* (different portions) was extracted twice with 20 volumes of water, and the 11αOH-KA content was measured (right). The means and SD are indicated, *n* = 3. (**B**) LC-UV analysis of *A. lavenia* leaves extracted with 10 volumes (left) and 50 volumes (right) of water. Arrows indicate 11αOH-KA. (**C**) One gram of dried *A. lavenia* leaf was extracted once with the indicated volumes of water, and the recovery of 11αOH-KA was calculated after the LC-UV analyses (*n* = 2). (**D**) Extracts (30 volumes of water) were incubated at the indicated temperatures and times, with the analyses performed by LC-UV. * and ** indicate *p* < 0.05 and *p* < 0.01, respectively (*n = 3*).

**Figure 4 foods-09-00073-f004:**
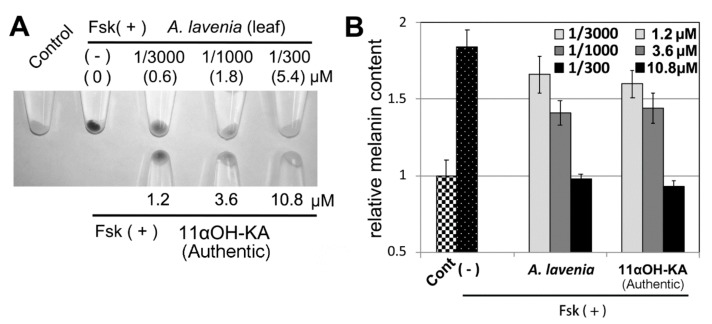
(**A**) *A. lavenia* leaves were extracted with 30 volumes of water and incubated at 120 °C for 20 min. After measurement of the 11αOH-KA content in the extract, the extracts (upper: actual concentrations of 11αOH-KA from the extract are indicated in brackets) and authentic 11αOH-KA with a 1/10–10-fold concentration were subjected to a melanogenesis inhibitory assay in the B16F10 cells. By visual examination, we selected sets of cells showing the same levels of melanin content (lower). (**B**) The melanin content in the **A:** extract and authentic samples was thus measured. The means and SD are indicted, *n* = 3.

**Figure 5 foods-09-00073-f005:**
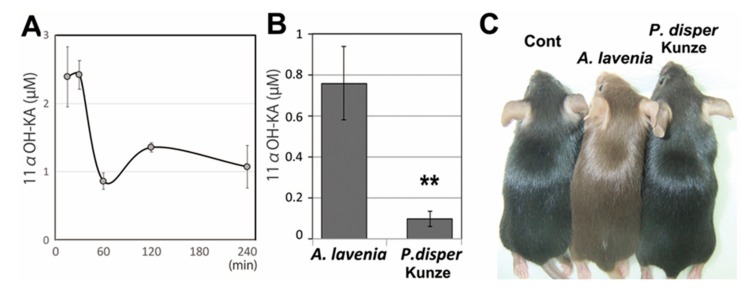
(**A**) Mice were orally administrated 11αOH-KA (100 mg/kg), and their blood was collected at the indicated time. The 11αOH-KA contents were measured by LC-MS analyses. The means and SD are indicted (*n* = 3). (**B**) Mice were treated with *A. lavenia* or *P. dispar* Kunze leaf extracts in drinking water (a twofold dilution of the extract which was prepared by the same methods in Figure 4A) for one week, and blood was collected for 11αOH-KA measurement (*n* = 3). The means and SD are shown. ** indicates *p* < 0.01. (**C**) Three-week-old mice (*n* = 2; see also Appendix A) were treated with the water extract (in Figure 5B) for two weeks, and then the pigmentation in the newly grown hair was examined by photography.

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
