# Peer review of "The High Content of Ent-11α-hydroxy-15-oxo-kaur- 16-en-19-oic Acid in Adenostemma lavenia (L.) O. Kuntze Leaf Extract: With Preliminary in Vivo Assays"

_foods, 2020, doi:10.3390/foods9010073_

Round 1
Reviewer 1 Report
Comments to Authors:
Authors would add (in vivo) in the title according to journal instructions. The authors could define the abbreviations especially when mentioned for the first time. In line 33, the authors could mention the biological activity that made investigation for it. In line 48, the authors would refer to plant parts with the high level of 11αOH-KA. In the introduction section, the authors can enumerate different pharmacological activities of the plant additionally the aim of the work as well. The introduction is not informative enough, and the authors could rewrite it with clear outlines in which every sub-topic would be covered comprehensively. Authors should add the time of the plant collection and its identification. In melanogenesis assay, please clarify the concentrations of lavenia water extract used in the assay. In line 119, give the reason for using this concentration as the normal concentration is 4,5 mM from glucose. In the results line 159, the authors could describe how the contaminants were identified. In the analysis of 11αOH KA in mouse serum, please explain why using this dilution. In line 156, the authors could indicate the results of each dilution used. The authors would organize the figures and describe them clearly. Authors would check the style of writing references in accordance to journal instructions. The authors could benefit from the following reference in the introduction section: Kanwal, A. Jabbar Siddiqui, H. R. El-Seedi, S. G. Musharraf (2018): Two-Stage Mass Spectrometry Approach for the Analysis of Triterpenoid Glycosides in Fagonia indica. RSC Advances, 8(71), 41023-41031.
Author Response
Authors would add (in vivo) in the title according to journal instructions.
Thank you for your suggestion, the title is now:
High Content of Ent-11α-hydroxy-15-oxo-kaur-16-en- 19-oic acid in Adenostemma lavenia (L.) O. Kuntze Leaf Extract: in vivo anti-melanogenic activity
The authors could define the abbreviations especially when mentioned for the first time.
Thank you for your suggestion, I mentioned full letters as follow:
We use full name of kurenoic acid in Title as :
・Ent-11α-hydroxy-15-oxo-kaur-16-en- 19-oic acid
・CHCl3 was changed to chloroform
・cyclic AMP (cAMP)
・L-3,4-dihydroxyphenylalanine, L-Dopa,
・chromatography (LC)
・mass (MS)
In line 33, the authors could mention the biological activity that made investigation for it.
Thank you for your suggestion, we changed the abstract as follow:
Methods: Anti-melanogenic activities of water extracts of A. lavenia leaves and Pteris dispar Kunze (a cognate of P. semipinnata) leaves were compared in mouse B16F10 melanoma cells. The amount of 11αOH-KA was measured by using liquid chromatography spectrometry. C57BL/6J mice were treated with the water extract of A. lavenia leaf, and the blood concentration of 11αOH-KA was measured. The in vivo efficacy of the water extract of A. lavenia leaf was evaluated by anti-melanogenic activity by monitoring hair color.
In line 48, the authors would refer to plant parts with the high level of 11αOH-KA.
Thank you for your suggestion, we added “leaves of”.
In the introduction section, the authors can enumerate different pharmacological activities of the plant additionally the aim of the work as well. The introduction is not informative enough, and the authors could rewrite it with clear outlines in which every sub-topic would be covered comprehensively.
Thank you for your suggestion, we changed several parts as follow:
・The first paragraph was separated by a break.
・The structure of 11αOH-KA was added as Figure 1.
・(We mistaken positions (15th and 16th)
・Other revisions are indicated red words.
Authors should add the time of the plant collection and its identification.
Thank you for your suggestion, we added information in Materials and Methods.
In melanogenesis assay, please clarify the concentrations of lavenia water extract used in the assay.
Thank you for your suggestion, we added more information (1/300, 1/1000, 1/3000) in a figure legend (Fig.1). In addition, we described how much precipitants were produced from A. lavenia and P. dispar Kunze in the legend.
・(A) B16F10 cells were treated with water extracts of A. lavenia and P. dispar Kunze (prepared with 20 volumes of water) at the indicated fold dilution (1/300, 1/1000, 1/3000) in the medium. When the water extracts were dried, 1 mL extracts from A. lavenia and P. dispar Kunze dried leaves (50 mg each) produced approximately 9 mg and 12 mg precipitants, respectively. Thus, the 1/1000 dilution made 9 µg/mL and 12 µg/mL, respectively.
・Melanogenesis was induced by 20 µM of forskolin (Fsk) for 48 h.
・total dry weight of extracts → total dry weight of the final fractions (27.5 mg and 54.5 mg, respectively)
In materials and Methods, we added information as:
・For 20-, 30-, and 50-volume extractions, 50 mg, 33 mg, and 20 mg of dried leaf powder was extracted by 1 mL of distilled water.
In line 119, give the reason for using this concentration as the normal concentration is 4,5 mM from glucose.
Thank you for your suggestion, 5.5 mM D-glucose was not correct. 4.5 g/L is correct.
・(DMEM-High glucose [4.5 g/L D-glucose], WAKO, Kyoto, Japan)
In the results line 159, the authors could describe how the contaminants were identified. In the analysis of 11αOH KA in mouse serum, please explain why using this dilution.
Thank you for your suggestion.
We did not identify the contaminants. “contaminants” was not good explanation. Therefore, we changed the word to
・ “unknown compounds”
In line 156, the authors could indicate the results of each dilution used.
Thank you for your suggestion, we added more information.
・At 1/3000 dilution, A. lavenia extract decreased the melanin content by 69% of control (Fsk only), and P. disper Kunze extract did by 41%. At 1/1000 dilution, A. lavenia extract decreased the melanin content completely, while P. disper Kunze extract did by 60%.
The authors would organize the figures and describe them clearly.
Thank you for your suggestion, we added information in Fig.4 and in figure legends.
Authors would check the style of writing references in accordance to journal instructions
Thank you for your suggestion, we corrected Journal abbreviations.
The authors could benefit from the following reference in the introduction section:
Thank you for your suggestion, we cited the reference in materials and methods.
Finally, Thank you very much for your helpful suggestions.
Reviewer 2 Report
This manuscript describes the purification of 11αOH-KA from AdenostemmA. L. Kuntze leaf and its function on anti-melanogenesis in cells and mice. The A. lavenia leaf can be used for potential source of materials for skin whitening. The extensive editing of English language and style are required.
Specific comments:
1.Line 75-81. The authors should list the objectives of this study, not results.
2.Line 92. What’s the model of HPLC?
line 104-116. The detail of animal study should be described. How many animal in each group? Please cite the reference for measuring 11αOH-KA using LC/MS. Figure 2 B. It seemed that 10 vol water extraction obtained higher signal than 50 vol water extraction. Figure 3B didn’t consist with the Figure 3A data. You mentioned “ 11αOH-KA in the water extract contributed to almost 50% of anti-melanogenic activity….” While in Figure 3B,A. lavenia and 11αOH-KA showed similar effect. For the animal study, the sample size is too small. Each group only contained 2-3 mice. In the discussion part, authors should discuss more about anti-melanogenesis effect. You didn’t test the anti-inflammation effect, which is not related to your current study.
Author Response
1.Line 75-81. The authors should list the objectives of this study, not results.
Thank you for your suggestion, we clearly mentioned objectives as follow:
Therefore, in this study, by measuring 11αOH-KA content and evaluating the efficacy of anti-melanogenic activities, we examined the potential of the water extract of A. lavenia leaf as a new category of functional foods, internal cosmetics.
2.Line 92. What’s the model of HPLC?
Thank you for your suggestion, we described full name of the model as:
Waters Xevo G2-XS QTof. Because this was not HPLC, we described UPLC (Ultra High Performance Liquid Chromatograph).
Line 104-116. The detail of animal study should be described. How many animal in each group?
Thank you for your suggestion, we considered animal welfare first. Therefore, we used only two mice for the treatment. Fortunately, the two mice produced the same result of hypopigmentation. No black mice shows hypopigmentation, in general.
We mentioned this in Materials and Methods:
・These animal experiments were approved by Animal Experiment Committee of Gifu University (H30-041). We used a minimum number of mice complying with animal welfare.
Please cite the reference for measuring 11αOH-KA using LC/MS.
Thank you for your suggestion, measurement of 11αOH-KA may be first time. Several reports showed the methods for kaurenoic acid by using UPLC-MS. We cited these in Materials and Methods:
[17,18]
Figure 2 B. It seemed that 10 vol water extraction obtained higher signal than 50 vol water extraction.
Thank you for your suggestion, Yes, 10 vol water extraction show a high peak. However, 5 times of the peak of 50 vol water was higher than 10 vol water. Because the levels of 11αOH-KA was saturated in 10 vol or 20 vol water-extraction, an apparent discrepancy (New Fig. 3B) was observed.
In addition, the position of (A)~(E) in new figures were not appropriate, which might lead the discrepancy. Therefore, (A)~(E) was moved to the top(s)of description in the legends for new figures.
・e.g., Figure 4. (A) A. lavenia leaves were extracted with 30 volumes of water and incubated at 120°C for 20 min.
Figure 3B didn’t consist with the Figure 3A data. You mentioned “ 11αOH-KA in the water extract contributed to almost 50% of anti-melanogenic activity….” While in Figure 3B,A. lavenia and 11αOH-KA showed similar effect.
Thank you for your suggestion, the original descriptions mislead the interpretation(s). We corrected the descriptions in Results and figure legends.
Results:
The visual examination (Fig. 4A) and the measurement of melanin content (Fig. 4B) indicated that the water extracts that showed the same efficacy of anti-melanogenic activity with authentic 11αOH-KA contained only a half amount of 11αOH-KA.
Legend:
・Figure 4. (A) A. lavenia leaves were extracted with 30 volumes of water and incubated at 120°C for 20 min. After measurement of 11αOH-KA content in the extract, the extracts (upper: actual concentrations of 11αOH-KA from the extract were indicated in brackets) and authentic 11αOH-KA with 1/10- ~ 10-fold concentration was subjected to melanogenesis inhibitory assay in B16F10 cells. By visual examination, we selected sets of cells showing the same levels of melanin content (lower). (B) Melanin content in (A: extract and authentic) was measured. Means and s.d. are indicted (n = 3).
We added “Authentic” in new Fig.4
For the animal study, the sample size is too small. Each group only contained 2-3 mice.
Thank you for your suggestion, we considered animal welfare and our previous experiences (PLoS One 2011, 6, e26148.). We performed experiments that could provide obvious differences (observations). We know that the number (2-3) of mice is not enough. However, no black-mouse changes their hair color by foods. The two mice be the first case being suppressed their hair pigmentation by foods.
We described this in Materials and Methods as:
We used a minimum number of mice complying with animal welfare.
In the discussion part, authors should discuss more about anti-melanogenesis effect. You didn’t test the anti-inflammation effect, which is not related to your current study.
Thank you for your suggestion, we discuss more melanogenesis.
Thank you very much for your helpful suggestions.
Round 2
Reviewer 2 Report
Line 125. Since it is the first time publishing the method using QTOF, you have to give more detail about the method, including QTOF condition, column information, internal standard et al.
Line 274, change "HPLC" to "UPLC'.
Line 387-400. These two paragraphs discussed anti-cancer and anti-inflammation effect of 11α-OH KA, which are not related to your study.
For animal study, 2 mice per group is not acceptable. You have to repeat this study using more mice.
Author Response
detail about the method, including QTOF condition, column information, internal standard et al.
Thank you for your suggestion. We described QTOF condition. We added data of LC-MS analyses in Supplementary Figure 2.
To normalize extraction efficiency, 10 nmols of 18β‐Glycyrrhetinic acid (Kanto Kagaku) was added to the serums prior to the extraction.
To separate 11α-OH KA, a C18 ODS column (3.0 mm × 150 mm [5 µm]: RP-18 GP, Kanto Kagaku) and a gradient ranging from 35% acetonitrile/water to 100% acetonitrile in 10 min were applied. The MS conditions were following: Capillary (kV) 2.20, Sampling Core 45, Extraction Cone 4.0. Because we failed to detect fragments of 11α-OH KA by MS/MS analyses (Supplementary Fig. 2A), 11α-OH KA was quantified by only MS by monitoring the mass to charge (m/z) of 331.1879 with 0.01 path (Supplementary Fig. 2B).
Line 274, change "HPLC" to "UPLC'.
Thank you for your checking. This part is HPLC, not UPLC.
Line 387-400. These two paragraphs discussed anti-cancer and anti-inflammation effect of 11α-OH KA, which are not related to your study.
Thank you for your suggestion. We removed these paragraphs and information about cellular targets for anti-cancer and anti-inflammation.
For animal study, 2 mice per group is not acceptable. You have to repeat this study using more mice.
Thank you for your suggestion. We re-performed the experiments with 3 mice. This time, 11α-OH KA concentration was decreased after 30 min. Probably, time (afternoon) might affect the excretion of 11α-OH KA from the blood.
We changed the data Fig. 5A